# Flexible Green Ammonia Production Plants: Small-Scale Simulations Based on Energy Aspects

**Guillermo de la Hera, Gema Ruiz-Gutiérrez, Javier R. Viguri and Berta Galán ***

Department of Chemical and Process & Resource Engineering, University of Cantabria, 39005 Santander, Spain; guillermo.delahera@unican.es (G.d.l.H.); gema.ruiz@unican.es (G.R.-G.); vigurij@unican.es (J.R.V.)
* Correspondence: berta.galan@unican.es

**Abstract:** The conventional Haber–Bosch process (HBP) for $NH_3$ production results in $CO_2$ emissions of almost 400 Mt/y and is responsible for 1–2% of global energy consumption; furthermore, HBP requires large-scale industrial equipment. Green or e-ammonia produced with hydrogen from alkaline water electrolysis using renewable energy and nitrogen from the air is considered an alternative to fossil-fuel-based ammonia production. Small-scale plants with the distributed on-site production of e-ammonia will begin to supplant centralized manufacturing in a carbon-neutral framework due to its flexibility and agility. In this study, a flexible small-scale $NH_3$ plant is analyzed with respect to three steps—$H_2$ generation, air separation, and $NH_3$ synthesis—to understand if milder operating conditions can benefit the process. This study investigates the aspects of flexible small-scale $NH_3$ plants powered by alkaline electrolyzer units with three specific capacities: 1 MW, 5 MW, and 10 MW. The analysis is carried out through Aspen Plus V14 simulations, and the primary criteria for selecting the pressure, temperature, and number of reactors are based on the maximum ammonia conversion and minimum energy consumption. The results show that: (i) the plant can be operated across a wide range of process variables while maintaining low energy consumption and (ii) alkaline electrolysis is responsible for the majority of energy consumption, followed by the ammonia synthesis loop and the obtention of $N_2$, which is negligible.

**Keywords:** ammonia; small scale; distributed production; green hydrogen; renewable energy; Aspen Plus® simulation; energy consumption

## 1. Introduction

Ammonia ($NH_3$) is an essential chemical in modern society due to its crucial role in fertilizer production [1]. Furthermore, in the ongoing search for clean fuels and innovative energy storage technologies, $NH_3$ is attracting significant interest as an energy vector in the hydrogen economy [2], an electro-fuel (e-fuel), and an energy carrier due to its elevated hydrogen ($H_2$) content, which is essential for on-demand hydrogen production in fuel cell applications [3].

With lower energy intensities than $H_2$ for storage as liquid $NH_3$ and offering high stability in transportation and storage, $NH_3$ appears as one of the most promising liquid energy carriers. Ammonia transportation is particularly attractive as it is marketed worldwide and can, therefore, use existing infrastructure. Furthermore, all components in the large-scale ammonia supply chain are established and have a very high level of technological readiness [4]. Indeed, the notable energy density of liquid ammonia has facilitated its evolution as an energy storage medium to address the intermittency of renewable energy, which is typically lost as curtailed electrical energy.

At the EU level, the European Climate Law established the goal set out in the European Green Deal: the aim is for Europe's economy and society to become climate-neutral by 2050. The law also sets the intermediate "Objective 55" target of reducing net greenhouse gas (GHG) emissions by at least 55% by 2030 compared to 1990 levels. Today, transport

emissions represent around 25% of the EU's total greenhouse gas emissions. Therefore, ambitious changes in transport are required. In this context, the FuelEU Maritime Regulation will promote the uptake of renewable and low-carbon fuels through the establishment of a target for gradual reductions with respect to the annual average GHG intensity of the energy used onboard ships [5]. In line with this strategy, the International Maritime Organization (IMO) has set ambitious targets for reductions in GHG emissions with a predicted scenario wherein e-ammonia will be one of the main marine fuels in the future [6–8].

However, the traditional Haber–Bosch process (HBP) used for $NH_3$ production contributes nearly 400 Mt $CO_2$/y, accounting for 1.2% of global $CO_2$ emissions; the process is responsible for 1–2% of global energy consumption, and it is energy- and greenhouse-gas-intensive [9]. "Green" or e-ammonia produced with hydrogen from water electrolysis using renewable energy and nitrogen from air is considered an alternative to fossil fuels [10]. Currently, this presents a challenge as authors such as Baeyens et al. [11], Li et al. [12], and Deng et al. [13] have assessed various alternative methods for $H_2$ production, both via the steam reforming of fossil resources and exploring renewable routes, which are nearing maturity and can produce $H_2$ at lower costs than electrolysis. Furthermore, a more recent study by Deng et al. [14] analyzed the economic feasibility of the thermo-catalytic decomposition of $CH_4$ into $H_2$ and C, resulting in competitive $H_2$ costs compared to current methane steam reforming.

Renewable solar and wind energy exhibit inherent fluctuations, often generating peak energy outputs that do not align with peak demands both on a daily and seasonal basis. Hence, energy storage is imperative. For day–night operations, short-term or daily energy storage through batteries emerges as the most viable solution. Conversely, long-term energy storage primarily relies on power–chemical conversion methods, wherein electricity is transformed into hydrogen via electrolysis. While reconverting hydrogen back into electricity may entail efficiency losses and challenges with respect to safe storage and distribution, the conversion of hydrogen into hydrogen carriers, such as ammonia, presents an appealing alternative for long-term storage purposes.

The primary challenge lies in its high costs, primarily resulting from the fact that electricity is more expensive than natural gas or coal in most countries worldwide. It is argued that an electrically driven HBP ammonia revolution will rely on developing agile processes that align with geographically isolated and intermittent renewable energy sources. Such an agile system may involve using renewable energy to produce ammonia for fertilizers and fuel in order to meet local electrical power demands and produce hydrogen for energy storage. Although decentralized $NH_3$ production does not experience the benefits of the economics of scale, it does experience the benefits of improved efficiency and less transport and logistics losses and grants tighter control over the supply chain [15].

Moreover, the utilization of distributed ammonia production via a small-scale, simplified, and electrified HBP is anticipated to enhance and facilitate access to fertilizers in some developing countries, with associated social and economic benefits underpinning growth and development. As decentralized electricity infrastructure is established in rural and impoverished regions through renewable energy sources, the potential presented by ammonia in mitigating fluctuations would enable self-sufficient energy generation without reliance on fossil fuels. The flexibility of a small-scale electrified HBP is pivotal in this scenario, especially for distributed solar and wind energy systems. Ammpower [16] developed an independent green ammonia production process to produce up to 4 metric tons of high-purity anhydrous ammonia per day using only water, air, and electricity. $NH_3$ fabrication is carried out on-site using modular shipping containers for on-demand production that is suitable for agricultural fertilizers or industrial applications.

Up until the 1990s, in order to reduce costs, green ammonia production with electrolysis was typically operated at large scales (equivalent to about 165 MW), demonstrating technological implementation readiness. However, as explained above, there is gradual progress toward on-site decentralized $NH_3$, and as a consequence, small-scale green $NH_3$ production based on renewable energy has gained attention. In fact, in some cases, given

appropriate economic incentives, such as the implementation of a carbon tax or the adoption of on-site distributed e-ammonia production, small-scale plants are poised to gradually replace centralized manufacturing facilities. The main developments in ammonia synthesis are as follows: (i) scale-up for enhanced energy efficiencies and (ii) scale-down for smaller investments so that medium-size plants will be built less frequently in the near future.

Among the extensive bibliography related to green ammonia, Osman et al. [17] outlined methodologies for the design, simulation, and optimization of an industrial-scale facility (1840 t/day) that is powered exclusively by renewable electricity. The plant achieves a specific energy consumption of 10.43 kWh/kg-NH$_3$, obtaining 37.4% electric efficiency. Using linear optimization, the optimal generation and storage configuration for operating this plant is determined, considering hourly resolution operations and flexible sub-process options for operation at high and low capacities. More recently, Olabi et al. [18] summarized most emerging methods for direct ammonia synthesis and concluded that all synthesis routes still have considerable challenges, such as low efficiencies, high costs, and negative environmental impacts. David et al. [19] presented the "2023 Roadmap on Ammonia as a Carbon-Free Fuel", which provides a comprehensive assessment of the current worldwide ammonia landscape, future opportunities, and associated challenges facing the use of ammonia.

Focusing on a small-scale bibliography, Lin et al. [20] carried out a techno-economic analysis of a 20,000 t/year green NH$_3$ production facility. The investigation explores two different configurations of the Haber−Bosch process: high-pressure reaction–condensation and low-pressure reaction–absorption. The study suggests that the low-pressure process could be employed for the thermochemical energy storage of renewable resources in scenarios where small ammonia plants could be implemented. The authors demonstrate that, under certain conditions, small-scale all-electric ammonia production can become economically viable. Similarly, Patil et al. [21] demonstrated that the decentralized production of ammonia from renewables is achievable through Proton's NFuel small-scale unit, which is capable of producing 120 kg/h NH$_3$. Ammonia can be converted into power when needed via ammonia generators. Sousa et al. [10] investigated the feasibility of obtaining 25,000 t/year of green NH$_3$ (30 MW ammonia plant) as an energy carrier using green H$_2$ produced from a 1 MW PEM electrolyzer powered by a hydropower pant; the authors conclude that small-scale NH$_3$ generation based on hydropower remains economically uncompetitive in comparison to conventional NH$_3$ production methods. Cardoso et al. [22] delivered a techno-economic evaluation of a small-scale (1 MW) green ammonia production facility using a biomass gasification power plant. Under the studied market conditions, the power plant is predictably economically viable; additionally, the chosen geographic location for the biomass–ammonia power plant has the potential to facilitate a bioeconomy and circular economy framework.

Other small-scale references studied interesting aspects, such as long-term effects or intensification. Rouwenhorst et al. [23] provided a review focused on islanded ammonia economies, with particular emphasis on long-term feasibility within the size range of 1 to 10 MW; this work includes a comparison of electricity consumption and other characteristics of several small-scale ammonia synthesis plants. The electricity consumption of the proposed power-to-ammonia processes ranges from 8.7 to 10.3 kWh/kg NH$_3$, comparable to that of a large-scale Haber–Bosch process with a low-temperature PEM electrolyzer (8.6–9.5 kWh/kg NH$_3$). Spatolisano et al. [24] analyzed the process intensification of NH$_3$ production, introducing an NH$_3$ removal stage via absorption with a phosphate solution downstream of the reaction stage. The adoption of absorption for ammonia separation could potentially enable a moderate pressure synthesis loop (200 bar) and may facilitate the development of a down-scalable process based on renewable energy sources. The authors emphasized the importance of high energy efficiency and flexibility in transitioning from large-scale concentrated synthesis to small-scale distributed NH$_3$ synthesis. Koschwitz et al. [25,26] introduced a small-scale containerized power-to-ammonia system designed for remote areas, serving as an affordable and rapidly reacting chemical energy storage

solution for fluctuating renewable energy sources. Hydrogen, generated by an electrolyzer with a maximum capacity of 15 kW, serves as the inlet for the system. The described operation boasts a zero-carbon footprint and a maximum daily ammonia production capacity of 35 kg. The authors conducted an exergy analysis to compare two different configurations of the small-scale 15 kW plant, identifying the optimal layout in terms of both economic and environmental considerations.

Ammonia synthesis using natural gas as feedstock is a mature technology, but the use of renewable electricity as feedstock still requires substantial research. The aim of the present work is to determine the operational conditions for different electrolyzer capacities to generate between 5 and 55 kmol/h of ammonia (plant powered by 1 to 10 MW electrolyzer capacity). The three steps required for "green" ammonia production, $H_2$ generation, air separation, and $NH_3$ synthesis are analyzed. Simulations of the integrated plant are carried out using the Aspen Plus® Version 14 software in order to obtain the appropriate technical conditions that result in a reduction in consumption.

## 2. Materials and Methods

### 2.1. System Boundary

Figure 1 shows different ammonia synthesis routes: the Haber–Bosch process and nitrogen reduction reaction (NRR) processes such as electrocatalysis, photocatalysis, photo-electrocatalysis, and the less studied alternatives, among which include plasma catalysis, bioelectrocatalysis, and catalysis via redox-mediated electrolytic nitrogen reduction reactions (RM-eNRR) [27,28]. In Figure 1, the processes for obtaining $N_2$ through air separation units (ASUs) consisting of membrane separation, PSA, or cryogenic distillation are shown. Additionally, various processes for obtaining $H_2$ are depicted, such as different types of water electrolysis, biomass gasification or reforming, and the steam reforming of natural gas; the latter two involve a subsequent $CO_2$ removal process. The raw materials for obtaining ammonia ($N_2$ and $H_2$) feed into different variants of the thermocatalytic Haber–Bosch process. Additionally, ammonia can be obtained via nitrogen reduction using electrochemical or photochemical processes. While these pathways for ammonia synthesis may potentially emerge as commercial technologies after 2030, enhancements to the existing Haber–Bosch process are likely to serve as the near-term alternative for sustainable ammonia synthesis [29].

The Haber–Bosch process, both the conventional process and the improved process with integration and intensification [15,30,31], involves the use of $N_2$ and $H_2$ as raw materials. The use of $H_2$ obtained through the electrolysis of water with renewable energies (e-hydrogen) gives rise to e-ammonia. In recent years, substantial efforts and advancements have been made in the research and development of green hydrogen production technologies, leading to the creation of various methods and technologies aimed at enhancing efficiency and cost reduction [32], such as the techniques of alkaline electrolysis (AEC), proton exchange membrane (PEM) electrolysis, and solid-oxide electrolysis (SOEC). The technologies used commercially to obtain nitrogen from air constitute the air separation unit and comprise cryogenic distillation, pressure swing adsorption, and membrane separation. Traditional paradigms of $NH_3$ process design in the chemical industry need to be re-optimized for the novel constraints of renewable energy; thus, this study focuses on the analysis of small-scale green ammonia plants when renewable energy is used. A small-scale ammonia production plant with nitrogen delivered through an air separation membrane, hydrogen produced via AEC water electrolysis, and two different configurations of Haber–Bosch converters is proposed in Figure 1.

The heat generated in the converter is recovered and transformed into thermal energy. The ammonia synthesis conversion rate is limited by temperature. At industrial scales, around 10–15% per pass, conversion is typically obtained in the reactor [33]; however, unconverted reactants can be separated from ammonia and recycled; thus, overall conversion can eventually reach 97% [34].

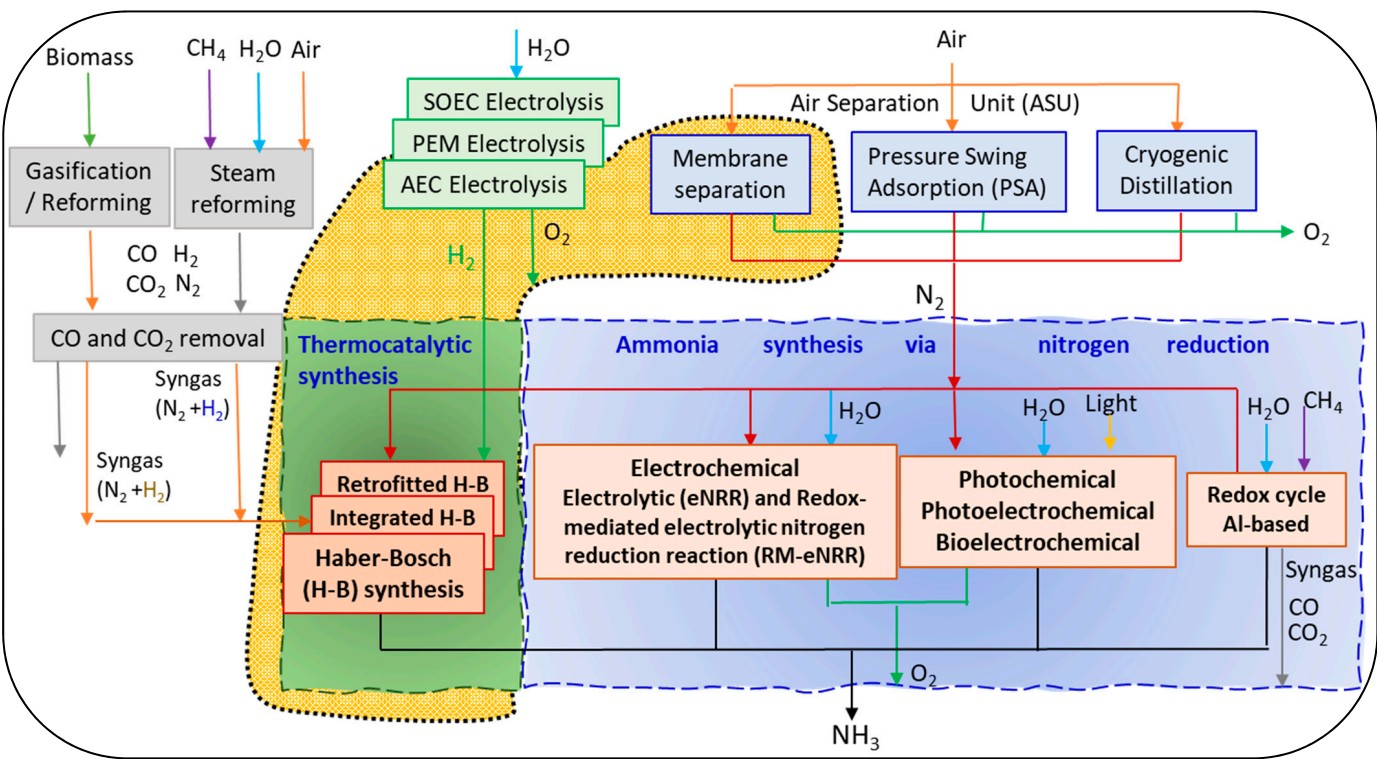

**Figure 1.** Ammonia synthesis routes and boundary of the studied system ( ). Studied processes in the present work.

In this work, the technological $NH_3$ production process is divided into three sections: the membrane air separation unit, $H_2$ generation with a base on an alkaline electrolyzer, and $NH_3$ synthesis considering a kinetic approach for the HB loop. Figure 2 shows a detailed flowchart of the complete process used in this work, highlighting each of the studied sections. Figure 2 illustrates the three process units involved in green ammonia production. The membrane air separation unit comprises three membrane modules (MODs) supplied with fresh and recirculated air, which is previously compressed (C) and cooled (HX). Oxygen and recirculated air permeate are obtained after vacuum application (VP); oxygen and nitrogen output streams are obtained from the ASU unit. In the electrolysis unit, water, along with liquid recirculation from the flash separator (F), is fed to alkaline electrolysis equipment (AEC) via pumps (PUMPs) and heat exchangers (HXs); purge and oxygen output streams, together with hydrogen output streams, are obtained in the electrolysis unit. The obtained $N_2$ and $H_2$ are mixed before entering the Haber–Bosch ammonia synthesis unit after compression (MCOMP and C equipment) and temperature increases (HX). The synthesis gas is fed to reactors (Rs), resulting in a stream that needs to be cooled (HX), separated through flash equipment (F), and partially recirculated back to the reactor; purge output streams and ammonia products are obtained from the synthesis unit.

### 2.2. Membrane Air Separation Unit

For small-scale systems, membrane permeation stands out as the preferred alternative for nitrogen production [28,35]. A typical gas separation membrane system consists of one or more membrane stages, with one or more compressors used to pressurize the feed streams and/or vacuum pumps to apply vacuum to the sides of the permeate [36]. Therefore, designing a process that uses the minimum number of stages to achieve fixed purity is advised.

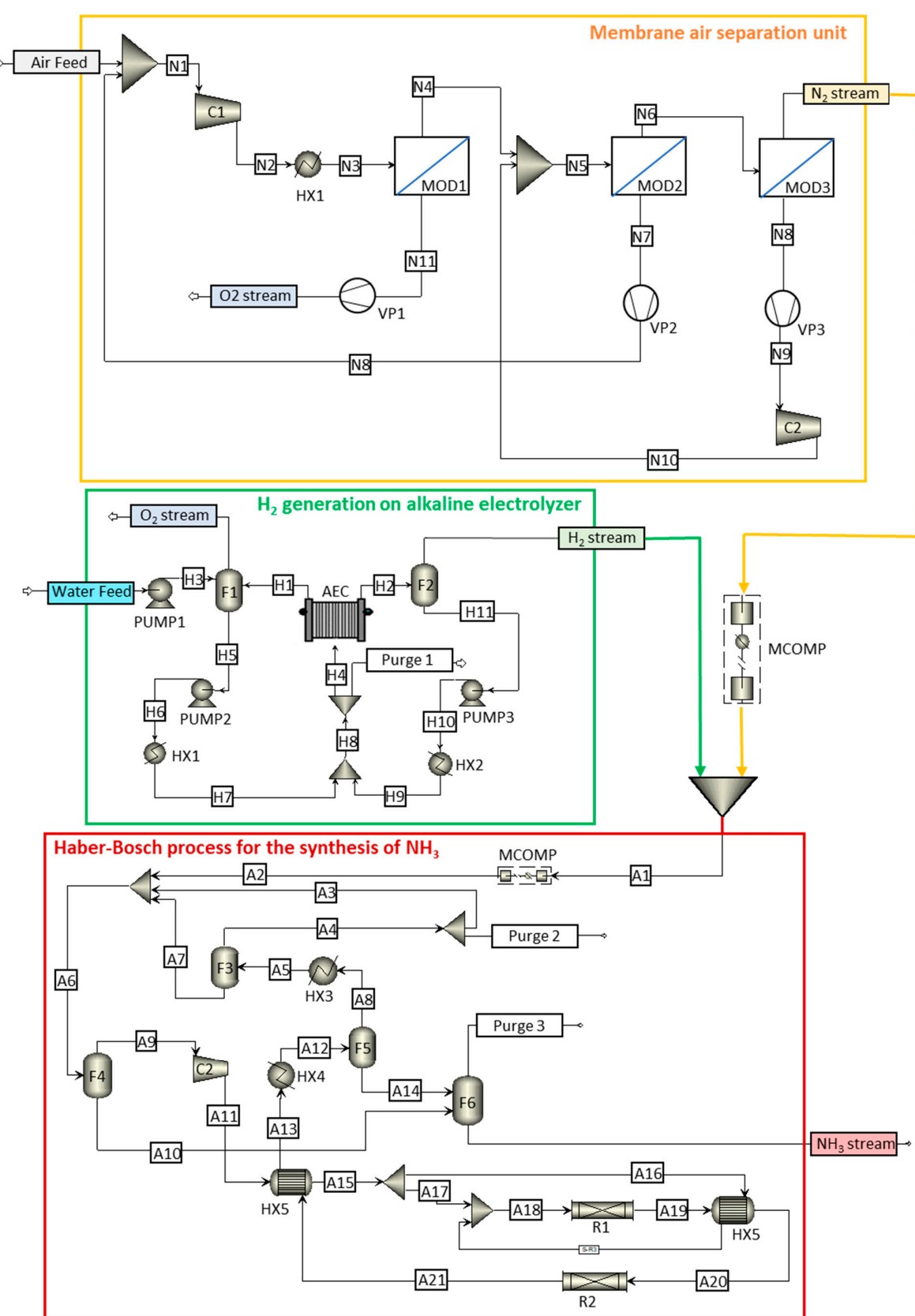

**Figure 2.** General process flowchart: yellow box shows the membrane air separation unit, green box shows the generation of $H_2$ in an alkaline electrolyzer, and red box shows the Haber–Bosch process for the synthesis of $NH_3$.

Bozorg et al. [37] analyzed the cost of nitrogen production from the air for four different levels of purity (90, 95, 99, and 99.9%) and identified the minimum cost and optimal process configuration with respect to currently commercially available membrane materials (the number of stages and whether to employ compression or vacuum operation and multistage configuration). They proposed an optimal configuration with three membrane modules to achieve a purity of 99.9%. Similarly, Adhikari et al. [38] also proposed a configuration with three membrane modules based on a techno-economic analysis of oxygen–nitrogen separation.

Consequently, to obtain high-purity nitrogen, a suitable configuration is shown in the yellow box in Figure 2. The air is compressed and heated before feeding the first of the membrane modules (MOD1). The permeate output stream of this module consists of an oxygen-enriched stream. The output of the retentate feeds the second membrane module (MOD2), for which its output feeds the third membrane module (MOD3); here, high-purity nitrogen (99.9% molar) is obtained as a product. The permeate outlets from the second and third membrane modules are recirculated to the previous membrane module after compression. The vacuum pumps shown in Figure 2 allow the pressure gradient to be maintained between the sides of the membrane, ensuring oxygen permeation through the membrane.

Using the Aspen Plus V14 Simulator, the compressors in Figure 2 are modeled as isentropic and provide an outlet pressure of 2.90 bar. The heat exchanger allows control of the air inlet temperature relative to the first module, which is 22 °C. Vacuum pumps allow for maintaining a pressure drop between both sides of the membrane at 0.20 bar. The membrane modules are simulated using the "Sep" block, which allows separation based on the fluxes of its components. In this work, these fluxes are based on the membrane area and permeability. Table 1 shows the characteristics of the commercial membrane used in the simulation [37].

**Table 1.** Membrane air separation unit characteristics.

| Characteristics | Values |
|---|---|
| Material | Polyphenylene oxide (PPO) |
| Membrane thickness ($\mu$m) | 1 |
| $O_2$ permeability (GPU) | 200 |
| $N_2$ permeability (GPU) | 44 |
| Selectivity ($\alpha_{O_2/N_2}$) | 4.5 |

*2.3. $H_2$ Generation on Alkaline Electrolyzer*

At the moment, different $H_2$ renewable production methods are feasible [39]. Biomethane and biomass gasification have been considered as alternative low-carbon $H_2$ sources. Taking into account the strengths, weaknesses, opportunities, and threats of the available electrolysis technologies and considering small-scale systems, alkaline electrolysis is the alternative selected in this work for hydrogen production [16,40,41]. The flow diagram of the $H_2$ generation process in an alkaline electrolyzer is shown in the green box in Figure 2. A stream of water and 35% potassium hydroxide is introduced into the alkaline electrolyzer, obtaining two output streams: one from the cathode (hydrogen) and the other from the anode (oxygen). The potassium hydroxide solution is recirculated, and the amount of water consumed in the electrolysis process is replaced. Later, $O_2$ and $H_2$ are purified so that pure oxygen and hydrogen gas (>99.5% purity) are obtained. The liquid streams from flash equipment are recirculated into the process. Purge 1 is used as a safety purge to maintain a constant inflow into the electrolyzer.

For the "electrolyzer" block, physical model equipment available in Aspen Plus V14 was used. It was designed with the characteristics shown in Table 2. These have been completed with the data from [42,43] and with data from the technical data sheets of commercial electrolyzers.

**Table 2.** Geometrical and electrical parameters of the electrolyzer (AEC) and the membrane.

| Parameter | AEC | Membrane |
|---|---|---|
| Active area (sqm) | 1.01 | --- |
| Channel width (m) | 0.750 | --- |
| Length of channel (m) | 1.35 | --- |
| Reference exchange current density (mA/sqcm) | 100 | --- |
| Resistivity (ohm-cm) | 2.00 | --- |
| Thickness (m) | --- | 0.000500 |
| Porosity | --- | 0.500 |
| Tortuosity | --- | 3.14 |

*2.4. Haber–Bosch Process for the Synthesis of Ammonia*

Conventional $NH_3$ production in the industry occurs via a synthesis loop known as the Haber−Bosch process. In this process, hydrogen gas reacts with nitrogen gas at a molar ratio of 3:1, under high temperatures and pressure, typically utilizing an iron catalyst, as described by Equation (1).

$$N_2(g) + 3H_2(g) \rightleftarrows 2NH_3(g) + \Delta H \qquad (1)$$

The reaction occurs at the catalyst's surface where nitrogen and hydrogen are consumed, leading to the formation of ammonia in an exothermic reaction. High temperatures result in fast kinetics but affect the conversion at equilibrium; low temperatures favor equilibrium, slowing down reaction kinetics and compromising $NH_3$ yields. Therefore, high inlet temperatures are necessary in order to achieve high reaction rates; however, a low outlet temperature is also necessary to achieve high equilibrium conversion. A trade-off between kinetics and equilibrium is accomplished through the use of temperatures between 320 and 550 °C and pressures between 150 and 300 bars [44]. Sagel et al. [45] indicate that for a relatively small-scale process, it is beneficial to work at moderate process conditions: 350–525 °C and 100–200 bar. Employing multiple catalyst beds arranged in series along with an intercooler system is a method that is utilized in the chemical industry to reconcile the trade-off between thermodynamics and kinetics. In the present work, the reactors have been modeled using an "RPLUG" block in Aspen Plus V14, and the kinetic expression proposed by Anders Nielsen (1968) [46] is utilized. It is noteworthy that in order to increase the efficiency of the Haber–Bosch process, Ruthenium (Ru), promoted with metallic alkali systems, has emerged as a second-generation catalyst because of its high activity and outstanding thermal stability. Ru exhibits more than 20 times higher activity compared to typical Ni-based catalysts with a broader range of $H_2$ and $N_2$ ratios at the same temperature and half the pressure [23,47,48]. Authors such as Smith and Torrente-Murciano L. [49], among others, overcame the limitations of conventional ammonia catalysts by reaching equilibrium with fast kinetics at moderate pressure (20 bar) and low temperatures (<300 °C), working with a ruthenium-based catalyst designed using nanostructured ceria as support and cesium as the electronic promoter.

A regular HB plant comprises five main elements: (1) a compressor that pressurizes the synthesis loop; (2) a reactor to convert $H_2$ and $N_2$ gas into $NH_3$; (3) a heat exchanger to heat the inlet stream of the reactor with heat from the exothermic ammonia synthesis reaction; (4) a separation system (via condensation or adsorption) to extract the ammonia from the process: the traditional high-pressure HB process is normally equipped with condensation for efficiency reasons while low-pressure synthesis makes use of absorption and adsorption; and (5) recycle and purge streams to recover the unreacted $H_2$ and $N_2$ gas in the reactor and remove inert gases, avoiding build-ups in the synthesis loop. All these elements appear in the red box in Figure 2: hydrogen and nitrogen streams are fed into a mixer, and the mixed stream later passes through a compressor and several heat exchangers to increase the pressure and temperature up to the operating conditions of the synthesis reactors. After synthesis, the ammonia reactor output stream is cooled and passes

through the separation section where part of the gas stream is recirculated into the process, and the liquid output is the final $NH_3$ product, with a purity greater than 99.6%.

## 3. Results and Discussion

In this work, a small-scale $NH_3$ plant is analyzed, and the following three steps are considered: $N_2$ production from the membrane unit (Section 3.1), $H_2$ generation from the alkaline electrolyzer (Section 3.2), and $NH_3$ synthesis (Section 3.3).

### 3.1. $N_2$ Production from the Membrane Unit: Operational Conditions

Ammonia production requires abundant atmospheric nitrogen ($N_2$). A configuration with three membrane modules in series (Figure 2) allows the obtainment of a nitrogen stream with a purity of 99.9%. The membrane areas of the three modules depend on the inlet's airflow, which will depend on the $N_2$ flow necessary to obtain ammonia. If an electrolysis capacity of 1 MW is used, the necessary flow for 99.9% $N_2$ purity is 3.06 kmol $N_2$/h, 15.3 kmol $N_2$/h for an electrolysis capacity of 5 MW, and 30.6 kmol $N_2$/h for an electrolysis capacity of 10 MW.

The permeate flux through the membrane for component i can be determined via Equation (2), considering the specifications shown in Table 3:

$$F_i = Per_i \, A_m \, \Delta P \qquad (2)$$

where $F_i$ represents the flux of component i in mol/s, $Per_i$ denotes the permeation of component i in mol m$^{-2}$ s$^{-1}$ Pa$^{-1}$, $A_m$ denotes the membrane area in m$^2$, and $\Delta P$ denotes the pressure drop between the sides of the membrane. The membrane areas of each module and the nitrogen fluxes that cross the membrane for the cases studied are shown in Table 3. The permeate flow leaving each module is determined by Equation (2). This flow depends on the permeability values (Table 2) and the membrane area. The membrane area of each module has been estimated to ensure that the outlet flow from the third module achieves a nitrogen purity of 99.9% [44].

**Table 3.** Membrane areas and $N_2$ flows for 1 to 10 MW electrolyzers.

| Module | 1 MW Electrolyzer | | 5 MW Electrolyzer | | 10 MW Electrolyzer | |
|---|---|---|---|---|---|---|
| | Membrane Area (m$^2$) | N$_2$ Flow (mol/s) | Membrane Area (m$^2$) | N$_2$ Flow (mol/s) | Membrane Area (m$^2$) | N$_2$ Flow (mol/s) |
| 1 | 180 | 0.0530 | 900 | 0.265 | 1800 | 0.531 |
| 2 | 64.0 | 0.0188 | 386 | 0.114 | 772 | 0.228 |
| 3 | 198 | 0.0583 | 1310 | 0.385 | 2610 | 0.770 |

The simulation of this process generates two final output streams: one rich in nitrogen and the other rich in oxygen. The characteristics of the stream rich in nitrogen are as follows: temperature at 23.6 °C, pressure at 2.90 bar, and a molar fraction of 0.9999. Finally, the molar flows are 3.06 kmol/h, 15.3 kmol/h, and 30.6 kmol/h for 1 MW, 5 MW, and 10 MW electrolyzers, respectively. To obtain these molar flows, the energy consumption of 3.30 kWh/kmol$N_2$ or 1.77 kWh/kmol$NH_3$ is required.

### 3.2. $H_2$ Generation from Alkaline Electrolyzers: Operational Conditions and Energy Consumption

From commercial alkaline technology (MCLYZER, Hydrogenics, AEM Multicore, H-TEC SYSTEMS), gateway electrolyzers were selected in this work, and their specifications [50] were considered for simulations using 2–20 stacks and 85 bipolar alkaline electrolysis cells of 1012.5 cm$^2$.

Most studies use between 60 and 100 °C and 1–30 bar in the electrolyzer [42,43,51–55]. In this paper, different case studies have been developed to study the influence of pressure and temperature in the alkaline electrolysis process at differently sized plants. Initially, the influence of pressure and a temperature of 75 °C is studied, and the results for 1 MW are

shown in Figure 3a. It is observed that, with increasing pressure, hydrogen production increases, and purity also improves. Jang et al. [56] reported that high-pressure water electrolysis has a large advantage in obtaining high-purity hydrogen (over 99.9%) because the amount of saturated water vapor contained in the hydrogen gas decreased. Later, the influence of the temperature was studied considering pressure that is equal to 30 bar [51,54], and it was observed that by increasing the temperature in the electrolyzer, higher hydrogen fluxes are obtained and purity decreases (Figure 3b); however, all studied temperatures fulfill the purity requirements used in ammonia production since it is higher than 99.8%.

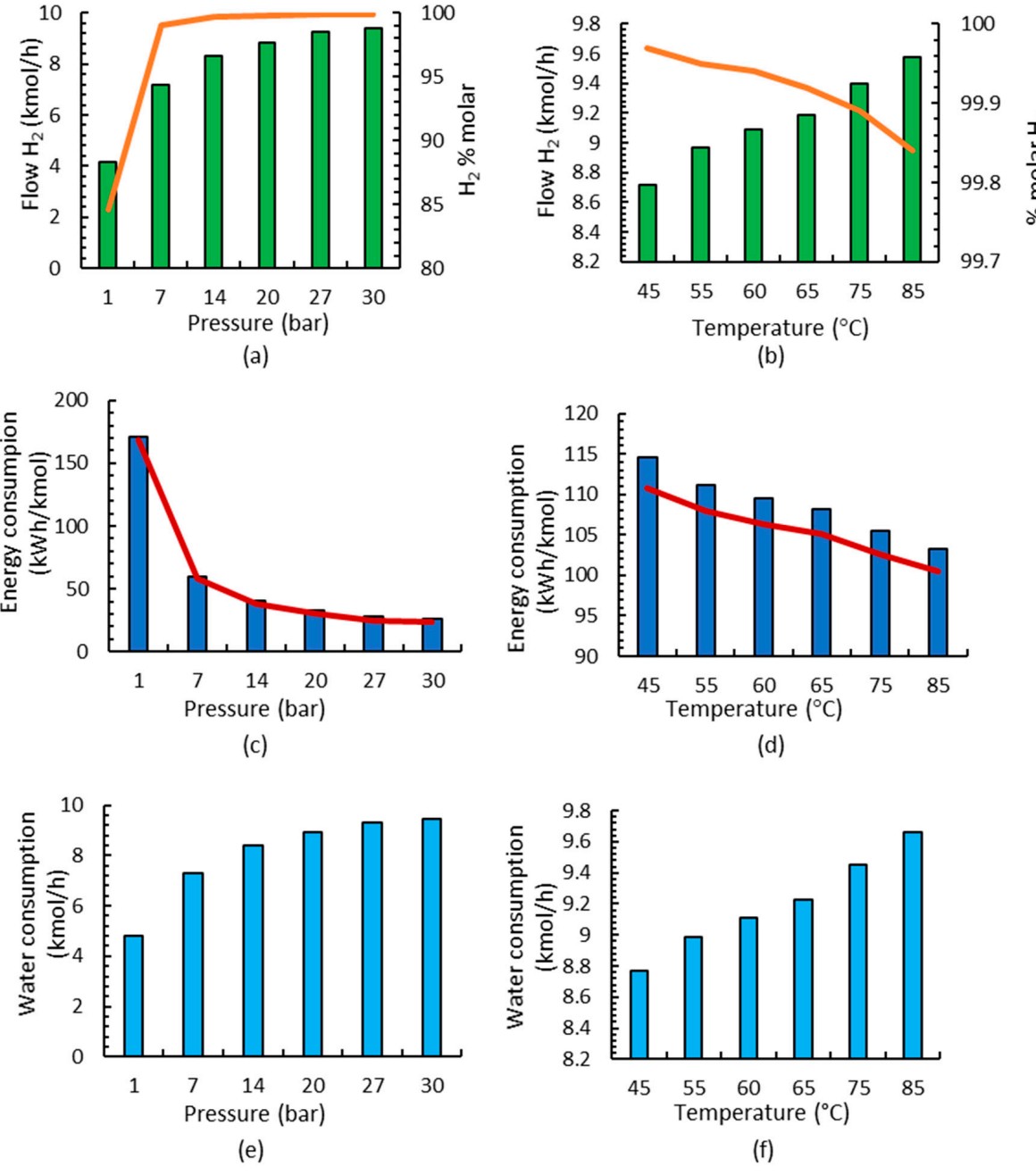

**Figure 3.** Results of the alkaline electrolysis stage for a 1 MW electrolyzer: (**a**) flow and purity of the $H_2$ stream working at different pressures; (**b**) flow and purity of the $H_2$ stream working at different temperatures; (**c**) energy consumption at different pressures; electrolyzer consumption shown by red line; (**d**) energy consumption at different temperatures; electrolyzer consumption shown by red line; (**e**) water consumption at different pressures; (**f**) water consumption at different temperatures.

The analysis of energy consumption is also carried out since the objective is to obtain the best technical conditions with the lowest energy consumption. In Figure 3c,d, it is observed that the influence of pressure on energy consumption is more important than the influence of temperature; it is observed that energy decreases linearly when increasing the temperature, but when increasing pressure, energy consumption is substantially higher. A more in-depth investigation of energy consumption shows that, in all cases, electrolyzer consumption corresponds to 97% of the total energy consumption and 3% to heat exchangers (2.9%) since pump consumption is negligible.

The influence of the consumed water is also analyzed for different pressures and temperatures in Figure 3e,f, observing the substantial influence of pressure and the small influence of temperatures relative to the $H_2$ production.

Lastly, the operating conditions of the alkaline electrolyzer were chosen based on the lowest energy consumption together with the higher flow and purity; a pressure of 30 bar and temperature of 60 °C were selected for the next production stages because these were the best conditions in the electrolyzer stage. Once the temperature and pressure have been selected, the small-scale production with respect to the 5 and 10 MW electrolyzers was analyzed, considering the same conditions based on the lowest energy consumption and highest $H_2$ production.

To produce 1 kmol of ammonia, 1.50 kmol of hydrogen needs to be produced. In this work, an electrical energy consumption rate of 4.86 kWh/Nm$^3$H$_2$ (N is "normal" in this context), 109 kWh/kmolH$_2$, or 175 kWh/kmolNH$_3$ is consumed under a pressure of 30 bar and 60 °C at small-scale plants using an alkaline electrolyzer. From the literature, gateway electrolyzers report 4.28 kWh/Nm$^3$H$_2$ of stack consumption [57].

Frattini et al. [57] reported an energy consumption rate of 231 kWh/kmolNH$_3$ for hydrogen production using an electrolyzer with Aspen Plus. More recently, Sousa et al. [10] reported 165 kWh/kmolNH$_3$ [57], and AmmPower [58] reported 149 kWh/kmolNH$_3$ of electric power.

### 3.3. HBP Ammonia Synthesis Loop: Operational Conditions and Energy Consumption

The HB process is well optimized for large scales; however, currently, wind and solar resources mainly support small-scale processes. In this section, the technical conditions of small-scale ammonia plants are analyzed considering the electrolyzer's capacity from 1 MW to 10 MW. The objective is to understand if "reduced" operating conditions in the process or small changes in regular configurations could benefit small-scale processes.

3.3.1. Influence of the Number of Reactors on NH$_3$ Conversion

For large-scale plants, a three-reactor system is the most efficient and cost-effective solution [57]. For small-scale plants, the number of reactors to be used needs to be analyzed. In this section, the influence of the number of reactors, as well as the influence of the reactor's pressure on ammonia conversion and energy consumption, is analyzed. In this case, the production of NH$_3$ is simulated using one to three separate adiabatic reactors, each representing a single bed. Figure 4 shows the NH$_3$ synthesis flow diagrams considering one, two, or three reactors. The output stream of each reactor is cooled down by a heat integration exchanger before entering the following reactor.

This study was carried out using reactors operating at 295 bar and considering a total pressure drop of 2, 3, and 5 bars when using 1, 2, or 3 reactors, respectively. These pressure drops along the reactor are negligible compared with the total pressure in the system. The length of each reactor is fixed (1 m) in order to analyze the influence of the number of reactors at different capacities. Normally, for accommodating the higher ammonia content and feed flow rate, reactors 2 and 3 are of a larger volume compared to reactor 1 [44]; however, in this work, the same volume is considered since different capacities are analyzed.

Firstly, the influence of the number of reactors on NH$_3$ conversion operating at an electrolyzer capacity of 1 MW can be observed in Figure 5. The results show that the

conversion increases up to 16.6% when using one reactor; however, the conversion increases to 23.8% when using two reactors, and when three reactors are used, the results are nearly the same as using two reactors (23.7%).

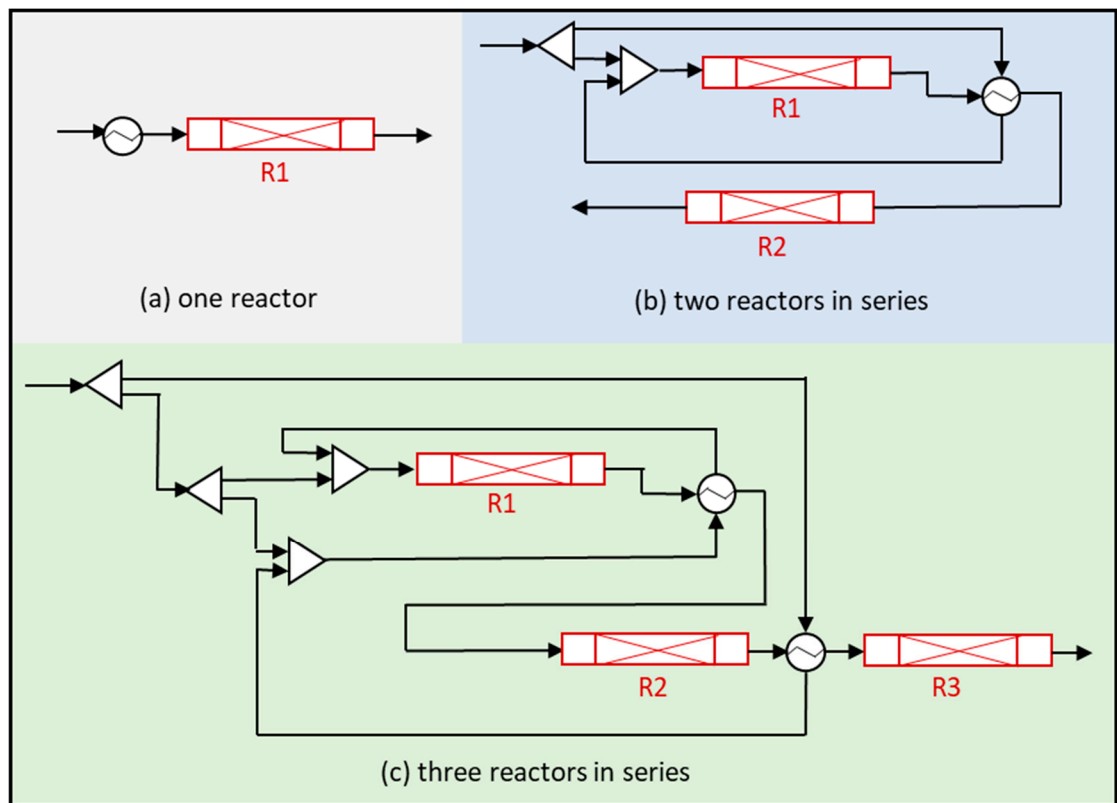

**Figure 4.** Flow diagram of $NH_3$ synthesis considering one (R1), two (R1 and R2), or three (R1, R2, and R3) reactors.

For the 5 MW capacity, it is observed in Figure 5 that the conversion is 16.5% when using a single reactor; when using two reactors, the conversion increases to 23.7%; when using three reactors, the conversion reaches 23.8%. It is clear that when using two or three reactors, the final conversion is similar. It is also observed that the first reactor achieves greater conversion (65%). Similar conversions are obtained at each reactor for 1 MW compared to 5 MW, but a reactor with a shorter length is required; furthermore, the final conversion is also equal at 1 and 5 MW (23.8%). From the results obtained with the 10 MW, it is observed (Figure 5) that the complete length of the first reactor is required to obtain 14.4% conversion. Using two reactors, the conversion increases up to 23.0%, and using three reactors, the conversion increases slightly up to 23.1%. Considering the final conversion, it can be inferred that it is necessary to work with two or three reactors to reach the highest conversion.

Figure 5 also shows the changes in temperature along the length of the reactor when working from 1 MW up to 10 MW electrolyzer capacities. Ammonia synthesis is an exothermic reaction that releases heat; therefore, the temperature along each reactor increases. The increase in reactant conversion and temperature occurs at a much higher rate in reactor 1 than in reactors 2 and 3 due to the low ammonia content. Figure 5 shows that in all cases, the highest temperature is obtained at the end of the first reactor because the highest increment of conversion is obtained at the first reactor. After each reactor is operated, there is a sharp decrease in temperature because the gas is cooled down by cold streams before entering the next reactor.

Finally, the number of reactors has no relevant influence on the purity of the ammonia product stream because it is higher than 99.6% in all cases, and the number of reactors has

a small influence on the total $NH_3$ flux, which varies from 5.60 ($\pm$0.1) kmol/h and 28.7 ($\pm$0.2) kmol/h to 56.7 ($\pm$0.5) kmol/h for capacities of 1, 5, and 10 MW, respectively.

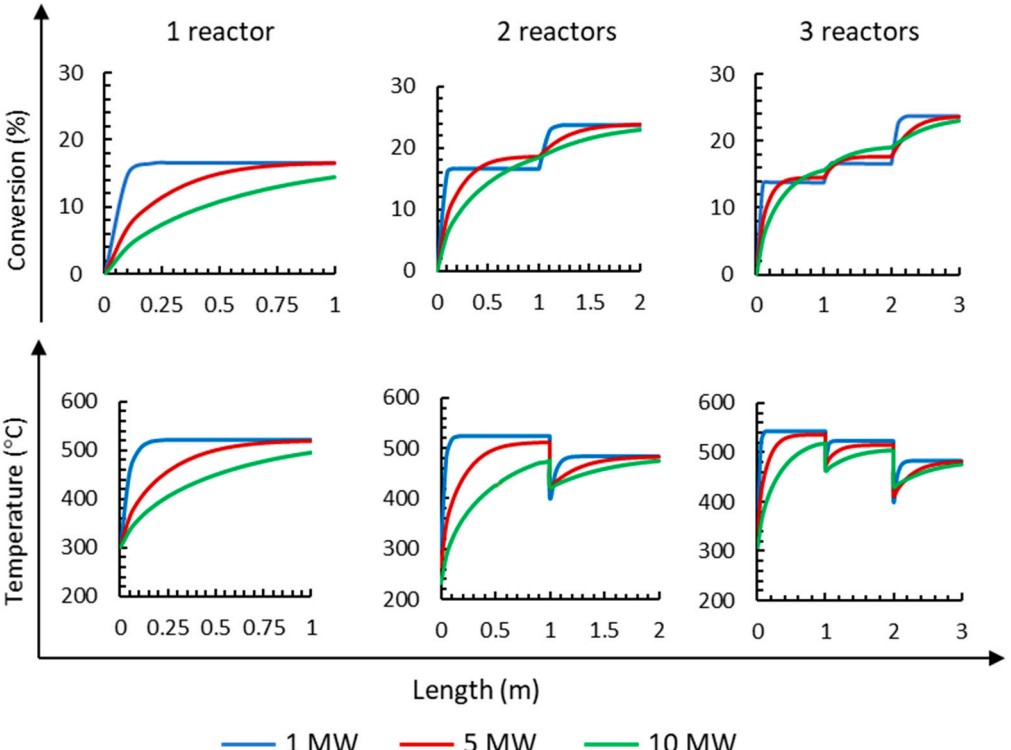

**Figure 5.** Influence of the number of reactors on $NH_3$ mol fraction, conversion, and temperatures for three different electrolyzer capacities of 1, 5, and 10 MW operating at 295 bar.

### 3.3.2. Influence of the Number of Reactors on $NH_3$ Consumption Energy

$NH_3$ production is energy-consuming as a result of the considerable compression operations necessary for the synthesis loop, the refrigeration loop, and the vapor−liquid separation at the end of the reaction loop. In this work, energy consumption is calculated based on the power requirements of the system, with data retrieved from the simulation carried out in Aspen Plus V14. The information in Figure 6a outlines a technical assessment of total energy consumption using one, two, or three reactors and different power levels from 1 MW to 10 MW relative to electrolyzer capacities. This information suggests that the use of a single reactor results in higher energy consumption due to increased liquid recirculation, which is a consequence of the lower total conversion compared to configurations involving two or three reactors; furthermore, the temperature at the reactor's outlet is higher than when using two reactors. Additionally, Figure 6a shows a small increase in energy consumption when employing three reactors compared to two, and this is attributed to the addition of an extra heat exchanger between reactor 2 and reactor 3.

It is also observed in Figure 6a that when working with two and three reactors, as expected, the larger plant is slightly more efficient than the smaller plant.

The information shown in Figures 5 and 6a outlines that a configuration of two reactors is the recommended option for small-scale ammonia production under working conditions of 295 bar and the three studied electrolyzer capacities. Therefore, the next sections of the paper investigate the two-reactor configuration.

Finally, Figure 6b shows a comparative analysis of the ammonia production processes' energy consumption by parts: recycled cooling, feed compression, and feed heat in the three different small-scale processes used in this paper. It is observed that recycled cooling is the main energy consumer, and this percentage increases with an increase in power. Feed

heat also requires high energy use that decreases when the power increases. Finally, the compression step is the least energy-demanding.

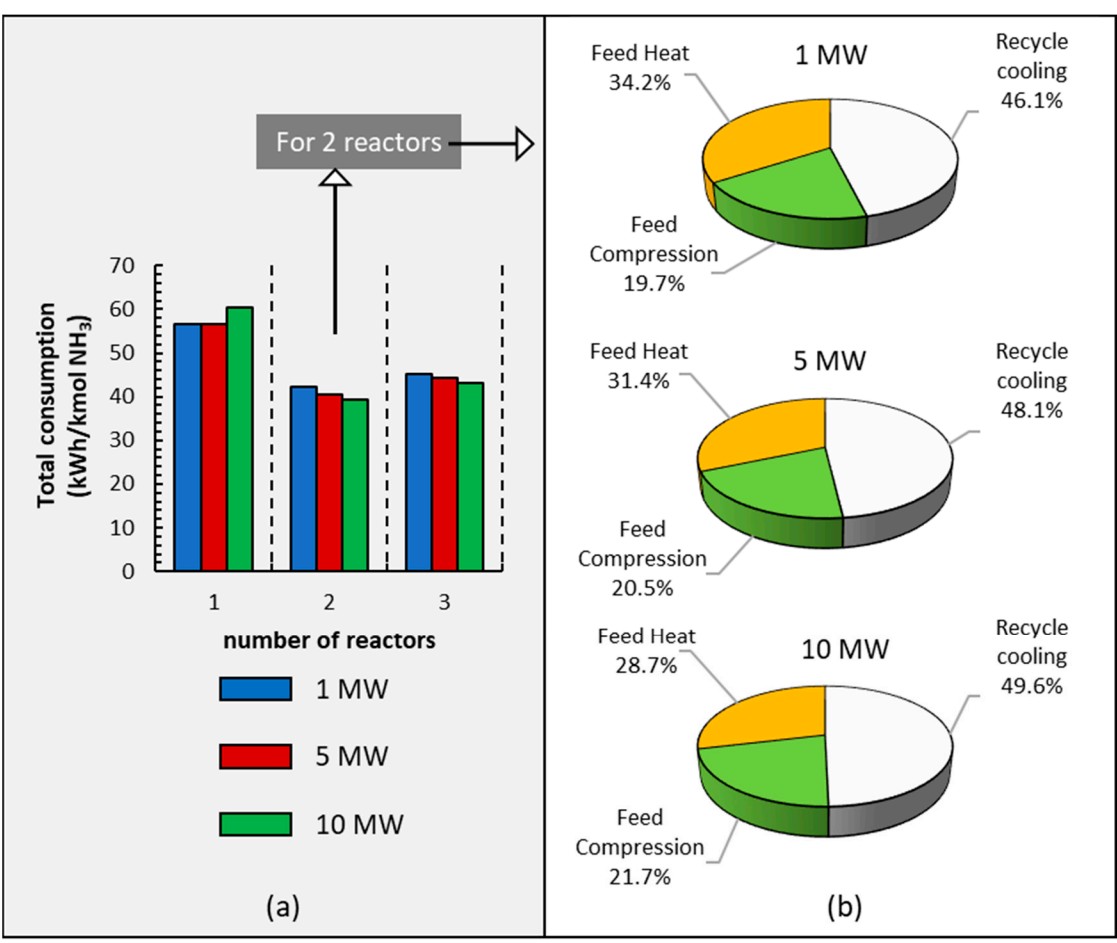

**Figure 6.** (**a**) Energy consumption using different numbers of reactors at different electrolyzer power capacities: 1, 5, and 10 MW. (**b**) Energy consumption in each section of the ammonia plant for the two reactors, with three different power capacities operating at 295 bar.

### 3.3.3. Influence of Pressure on Ammonia Conversion

Sagel et al. [45] indicated that moderate temperatures and pressures are preferred for small-scale ammonia production plants. Spatolisano et al. [24] indicated that when applied to small-scale production plants, $NH_3$ synthesis has to be performed at lower pressures in order to reduce operating costs, but low pressures negatively affect the chemical equilibrium of the reacting system; consequently, the authors used 200 bar.

Therefore, in this paper, the working pressure of reactors is analyzed, and four case studies are carried out at different pressures (160, 205, 250, and 295 bars) in order to outline the influence of pressure on $NH_3$ production and energy consumption. As in the previous sections, the influence of small-scale plants is also studied by analyzing three electrolyzers powered at 1, 5, and 10 MW.

Figure 7a shows that, in all cases, low pressures result in lower conversion because pressure affects the chemical equilibrium of the reacting system. It is also observed that $NH_3$ production is marginally influenced by pressure for each electrolyzer's power setting due to condensation and recycling loop conditions. Furthermore, purity is not influenced by pressure since it was higher than 99.6% in all cases.

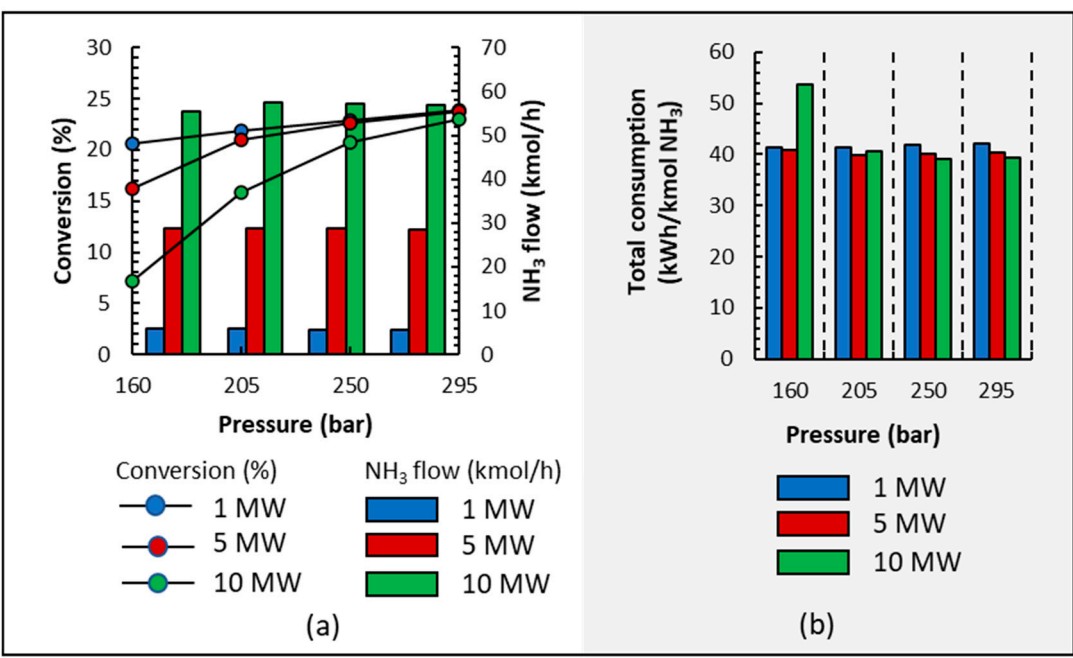

**Figure 7.** (**a**) Conversion and NH$_3$ output stream production; (**b**) energy consumption when operating at different pressures and electrolyzer capacities of 1, 5, and 10 MW.

### 3.3.4. Influence of Pressure on Energy Consumption

Figure 7b shows the energy consumption of NH$_3$ synthesis; in all cases, pressure has a small influence on energy consumption, except when operating at 160 bar and 10 MW where the highest conversion is 8% (see Figure 7a); therefore, the recycling loop is very energy-demanding. It is clear that lower pressures save energy from being consumed during compression but increase energy consumption during heating because conversion is lower and more liquid is recirculated; in short, the energy saved during compression is spent on heating and cooling. Even though there are slight differences, the energy consumed is higher when working at 295 bar than at 250 or 205 bars.

Traditionally, NH$_3$ production takes advantage of the economy of the scale, and it is observed that under the operation conditions represented in Figure 7b, a small effect of the economy scale is also observed because 1–3 kWh/kmol NH$_3$ is reduced when the plant's size is larger.

Taking into account previous results and the recommendation of operating at lower pressures due to safety considerations, the final recommended pressure varies between 205 and 250 bars, but it is mainly closer to 205 bar; this aspect is very important when carrying out small-scale processes that will mostly be operated by non-technical operators. In fact, Ammpower [16] commercializes two complete small-scale green ammonia plants inside shipping containers using three containers (20′/40′) for the smallest scale (0.3 MW) and seven containers (20′/40′) for another small-scale plant (1.7 MW). The Ammpower company recommends using these plants at farms and coops or in industrial applications. Other firms, such as FuelPositive [59] and Atmonia [60], among others, also developed modular and scalable units inside containers to facilitate small-scale green ammonia production.

It is important to highlight that the synthesis loop under the study conditions reported in Sections 2 and 3 is viable for different electrolyzer production capacities, as it can be operated within a wide range of process variables while maintaining low energy consumption.

### 3.4. Energy Consumption in a Complete HBP Ammonia Synthesis Loop

For global energy analyses, the following three stages are considered: air separation, H$_2$ generation, and NH$_3$ synthesis. Air separation energy represents the energy consumed

by the compressor and pump; $H_2$ generation energy represents the energy consumed by the electrolyzer and pumps; finally, $NH_3$ production energy is divided into two parts: feed compression energy, which embraces the compressors to increase the pressure of the $H_2$ and $N_2$ streams, and heat exchanger energy, which embraces the energy required for the refrigeration cycle and the energy required for feed heat. This analysis was carried out for 1, 5, and 10 MW powered electrolyzers using two reactors and operating at 205 and 250 bars. It is observed that this plant consumes between 214 and 216 kWh/kmol $NH_3$ when operating at 205 bar and consumes between 215 and 217 kWh/kmol $NH_3$ when operating at 250 bar. Section 3.3.4. recommends operations to be carried out between 205 and 250 bars due to safety considerations.

A Sankey diagram is shown in Figure 8 to visualize energy consumption at different stages of the $NH_3$ production process (hydrogen, nitrogen, and ammonia generation) relative to 5 MW of electrolysis power and operations at the recommended conditions of 205 bar. It is observed in Figure 8 that the electrolysis step requires the highest energy consumption (80.6%), followed by ammonia production (18.6%); in contrast, $N_2$ production exhibits very low energy consumption (0.818%). The results obtained at other capacities and 250 bar (not shown) result in similar energy demands because electrolysis is largely the step with the highest energy consumption.

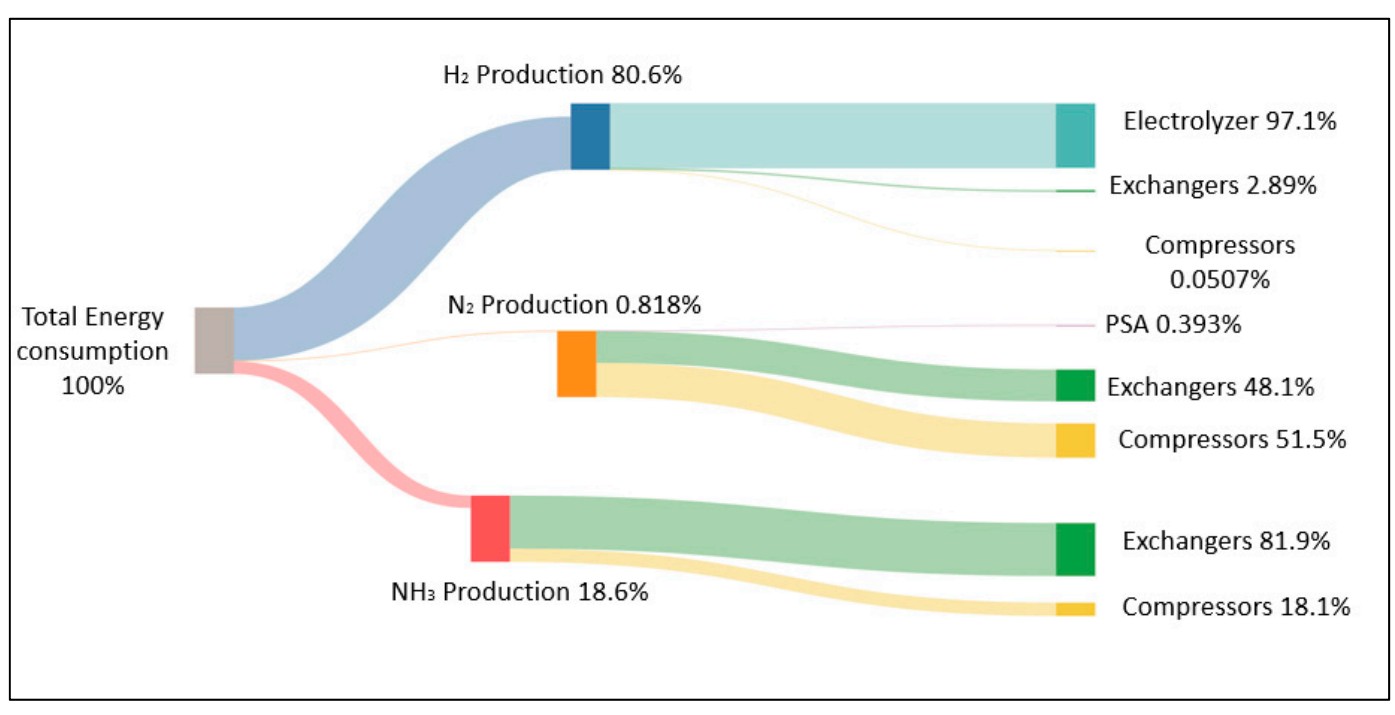

**Figure 8.** Sankey diagram for global energy analyses working at 5 MW capacity.

Ammonia production will always be an energy-intensive process. In fact, the Ammonia Energy Association [2] indicates that the best energy consumption values for natural gas, coal, and fuel oil are 132, 180, and 198 kWh/kmol $NH_3$, respectively (7.8, 10.6, and 11.6 kWh/kg $NH_3$); these values can increase depending on the region, plant size, and the source of hydrogen (feedstock). Furthermore, the corresponding $CO_2$ emissions of 1.6, 3.0, and 3.8 t $CO_2$/t $NH_3$ need to be considered. Making ammonia sustainably from renewable power is an even more energy-intensive process; in fact, Sousa et al. [10] indicate that green $NH_3$ production consumes about 2 kWh/kg $NH_3$ more than the conventional process.

Along the same line, other authors researching renewable ammonia synthesis technologies define an energy input range of 170–204 kWh/kmol $NH_3$ (9–12 kWh/kg $NH_3$), assuming that renewable energy (solar, tidal, or wind) is used to power an electrolyzer and an air separation unit, which fed $H_2$ and $N_2$ into a Haber–Bosch ammonia synthesis

loop [20,61]. Rouwenhorst et al. [23] reported different values for energy consumption depending on the scale: for large-scale plants, energy consumption varies between 9 and 11 kWh/kg $NH_3$, while energy consumption increases rapidly from 11 up to 22 kWh/kg relative to small-scale plants [23,61]. Other authors who work at small-scale production plants, such as Lin et al. [20], reported substantially lower values than the general bibliography: 165–158 kWh/kmol $NH_3$ (9.70–9.36 kWh/kg $NH_3$) at 30 MW when operating with high-pressure reaction–condensation and low-pressure reaction–absorption systems, respectively.

In relation to the energy consumption in each section of the ammonia plant, there are also different values in the literature: Sousa et al. [10] reported 11.0 kWh/kg $NH_3$ when working with an electrolyzer of 1 MW and reported that 88.5% of energy consumption comprises $H_2$ generation and 10.2% comprises $NH_3$ synthesis [45]. Lin et al. [20] reported very low consumption energies with respect to the $NH_3$ synthesis loop, which represents 9.1% of the total energy when working with high-pressure reaction–condensation and 3.5% when working with low-pressure reaction–absorption. Wiskich and Rapson [58] reported higher values for $H_2$ production (16.5%), which are similar to the values in Figure 8.

The differences in energy consumption observed in the literature are mainly based on ammonia synthesis plants: (i) the different working conditions in the reactors ($H_2$:$N_2$ ratio, inert gas concentration, pressure, and temperature); (ii) type of catalyst: conventional catalyst or advanced catalysts or even conventional iron-based catalyst of the newest generation; or (iii) the usage of adsorption concepts or conventional condensation at moderate temperatures, among others.

In relation to costs, although it is out of the scope of this paper, Smith and Torrente-Murciano [15] observed, across all their case studies, that the capital costs of renewable electricity and electrolyzers are the dominant process costs (70–80% of the cost at 100 MW) when ammonia is produced via the constant production HB process. Ishaq and Crawford [20] indicated that the costs of electrolytic ammonia production range from USD 680 to 900/t$NH_3$, which is anticipated to reach USD 400/t$NH_3$ by 2030. The use of cheaper techniques than electrolysis to produce the required hydrogen amount can reduce ammonia production costs. Baeyens et al. [62] carried out a comparative study of hydrogen production costs from renewable and non-renewable resources, highlighting that electrolysis options exhibit the highest unit cost (5.1–9.0 USD/kg $H_2$) in contrast to biomass-based options (1–2.8 USD/kg $H_2$); these latter techniques, along with nitrogen reduction processes, represent alternatives that could be considered in the future for the production of green ammonia.

## 4. Conclusions

In this study, a flexible small-scale $NH_3$ plant is analyzed considering the three stages—$N_2$ production from the membrane unit, $H_2$ generation from the alkaline electrolyzer, and iron-based catalyst Haber–Bosch $NH_3$ synthesis—for the final objective of analyzing the plant's operating conditions.

Aspen Plus simulations are carried out using alkaline electrolyzer units with three different capacities of 1 MW, 5 MW, and 10 MW. Firstly, the influence of electrolyzer pressure and temperature on energy consumption was studied, and it can be concluded that a pressure of 30 bar and a temperature of 60 °C are the best conditions. Later, the influence of the number and pressure at the $NH_3$ synthesis reactors is analyzed based on ammonia conversion and minimum energy consumption. The results show that a two-reactor configuration and a working pressure between 205 and 250 bars are the best conditions. An additional aspect contemplated in this study is that a lower operating pressure will result in an inherently safer design; furthermore, the fact that the small-scale processes will mostly be carried out by non-technical operators is also considered.

The results also show that alkaline electrolysis is notably responsible for the majority of energy consumption, followed by the ammonia synthesis loop, while energy for the obtention of $N_2$ is negligible.

From the results obtained in this paper, it can be concluded that the synthesis loop under study is viable for different production capacities, as it can be in operation within a wide range of process variables while maintaining low energy consumption.

The next step will be related to the optimization of the smallest plant powered by 1 MW and a production capacity of around 100 kg $NH_3$/h, which will be installed on a marine platform for a viability demonstration using offshore wind to supply the electrical power necessary for the plant.

**Author Contributions:** Conceptualization, J.R.V.; methodology, G.R.-G., G.d.l.H., B.G. and J.R.V.; validation, B.G. and G.R.-G.; formal analysis, J.R.V.; investigation, G.d.l.H., G.R.-G. and B.G.; writing— original draft preparation, B.G.; writing—review and editing, G.d.l.H., G.R.-G., B.G. and J.R.V.; visualization, B.G. and J.R.V.; supervision, J.R.V. All authors have read and agreed to the published version of the manuscript.

**Funding:** This research received no external funding.

**Data Availability Statement:** The original contributions presented in the study are included in this article, further inquiries can be directed to the corresponding author.

**Acknowledgments:** This study forms part of the ThinkInAzul programme and is supported by Ministerio de Ciencia e Innovación with funding from European Union Next Generation EU (PRTR-C17.I1) and by Comunidad Autónoma de Cantabria. Project: C17.I01—Plan Complementario de Ciencias Marinas.

**Conflicts of Interest:** The authors declare no conflicts of interest.

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
