# Peer review of "Flexible Green Ammonia Production Plants: Small-Scale Simulations Based on Energy Aspects"

_environments, doi:10.3390/environments11040071_

Round 1

Reviewer 1 Report

Comments and Suggestions for Authors

See attached comments. Major revision is recommended.

Comments on the Quality of English Language

Many language imperfections (lack of articles, some poor sentence structures, among others)/ To be corrected.

Reviewer 2 Report

Comments and Suggestions for Authors

Please respond to the following comments: 

1. it is recommended to add some findings to the abstract.

2. previous studies were not sufficiently described in the introduction section.

3. it is suggested to avoid the general description of the previous studies. It is recommended to rewrite the current references and add new references to this study.

4. it is recommended to put the aim of the paper at the end of the introduction.

5. why do low temperatures favor equilibrium and affect the reaction kinetics so that the NH3 yield decreases?

6. why does a configuration with three membrane modules in series (Figure 2) allow the recovery of a nitrogen stream with a purity of 99.9%?

7. why is it observed that increasing the pressure increases hydrogen production and improves purity?

8. why is the influence of the consumed water for different pressures and temperatures also analyzed in Figure 3 (e-f), where a large influence of pressure and a small influence of temperature is observed?

9. why do the number of reactors and the reactor pressure influence the ammonia conversion and the energy consumption?

10. the English language needs to be improved. 

Comments on the Quality of English Language

English language needs to be improved

Round 2

Reviewer 1 Report

Comments and Suggestions for Authors

Thanks for including modifications according to the initial review.

No further comments

Reviewer 2 Report

Comments and Suggestions for Authors

No comments. I have read the authors' answers and found that they correctly responded to all questions.